# Metabolic engineering considerations for the heterologous expression of xylose-catabolic pathways in *Saccharomyces cerevisiae*

Deokyeol Jeong[1☯], Eun Joong Oh[2☯], Ja Kyong Ko[3], Ju-Ock Nam[1], Hee-Soo Park[1], Yong-Su Jin[4,5], Eun Jung Lee[6]*, Soo Rin Kim[1]*

**1** School of Food Science and Biotechnology, Kyungpook National University, Daegu, Republic of Korea, **2** Renewable and Sustainable Energy Institute (RASEI), University of Colorado Boulder, Boulder, Colorado, United States of America, **3** Clean Energy Research Center, Korea Institute of Science and Technology (KIST), Seoul, Republic of Korea, **4** Carl R. Woese Institute for Genomic Biology, University of Illinois at Urbana-Champaign, Urbana, Illinois, United States of America, **5** Department of Food Science and Human Nutrition, University of Illinois at Urbana-Champaign, Urbana, Illinois, United States of America, **6** Department of Chemical Engineering, School of Applied Chemical Engineering, Kyungpook National University, Daegu, Republic of Korea

☯ These authors contributed equally to this work.
* eunjunglee@knu.ac.kr (EJL); soorinkim@knu.ac.kr (SRK)

**Data Availability Statement:** All relevant data are within the manuscript and its Supporting Information files.

**Funding:** This work was supported by grants from the National Research Foundation (NRF) of Korea

## Abstract

Xylose, the second most abundant sugar in lignocellulosic biomass hydrolysates, can be fermented by *Saccharomyces cerevisiae* expressing one of two heterologous xylose pathways: a xylose oxidoreductase pathway and a xylose isomerase pathway. Depending on the type of the pathway, its optimization strategies and the fermentation efficiencies vary significantly. In the present study, we constructed two isogenic strains expressing either the oxidoreductase pathway (XYL123) or the isomerase pathway (XI-XYL3), and delved into simple and reproducible ways to improve the resulting strains. First, the strains were subjected to the deletion of *PHO13*, overexpression of *TAL1*, and adaptive evolution, but those individual approaches were only effective in the XYL123 strain but not in the XI-XYL3 strain. Among other optimization strategies of the XI-XYL3 strain, we found that increasing the copy number of the xylose isomerase gene (*xylA*) is the most promising but yet preliminary strategy for the improvement. These results suggest that the oxidoreductase pathway might provide a simpler metabolic engineering strategy than the isomerase pathway for the development of efficient xylose-fermenting strains under the conditions tested in the present study.

## Introduction

Global climate change has accelerated efforts to find eco-friendly alternatives for fossil fuels. One idea is to use wood wastes and agricultural residues called lignocellulosic biomass, which does not interfere with food or the environment [1]. Lignocellulosic biomass, mainly composed of cellulose and hemicellulose, is hydrolyzed into glucose, xylose, and other simple and minor sugars, which can be transformed into biofuels and chemicals by microbial fermentation [2].

funded by the Korea government (NRF-2018R1A2B2007426). This work was also supported by the National Research Foundation of Korea (NRF) grant funded by the Korea government (MSIT) (No. 2019R1F1A1062633).

**Competing interests:** The authors have declared that no competing interests exist.

The yeast *Saccharomyces cerevisiae* is an industrial microorganism with superior sugar fermentation capabilities and stress tolerance. However, this yeast cannot metabolize xylose, requiring the introduction of a heterologous xylose pathway [3,4] as summarized in Fig 1A. The first step is to introduce either the NAD(P)H-specific xylose reductase/NAD$^+$-specific xylitol dehydrogenase (oxidoreductase, XR/XDH) pathway derived from *Pichia stipitis* or the xylose isomerase (XI) pathway derived from various anaerobic microorganisms, both of which convert xylose to xylulose. Next, xylulose is converted into xylulose-5-phosphate by xylulokinase either by endogenous but overexpressed *S. cerevisiae XKS1* or *P. stipitis XYL3*. Finally, xylulose-5-phosphate is metabolized into ethanol through the native pentose phosphate (PP) pathway connected to glycolysis in *S. cerevisiae*. In engineered strains of *S. cerevisiae* expressing the xylose oxidoreductase pathway, the rate of xylose consumption and ethanol productivity are relatively high, but xylitol, glycerol, and acetate are accumulated as byproducts [3,5]. This byproduct accumulation is mainly due to an unbalanced cofactor preference of the xylose oxidoreductase pathway, leading to a shortage of NAD$^+$ [3]. On the other hand, the xylose isomerase pathway is cofactor-independent, the expression of which in *S. cerevisiae* can lead to a high ethanol yield with minimal byproduct accumulation even under anaerobic conditions [6]. However, slow growth and xylose consumption were commonly observed in the engineered *S. cerevisiae* strains expressing the xylose isomerase pathway compared to those expressing xylose oxidoreductase pathway [5,7,8].

**(A)**

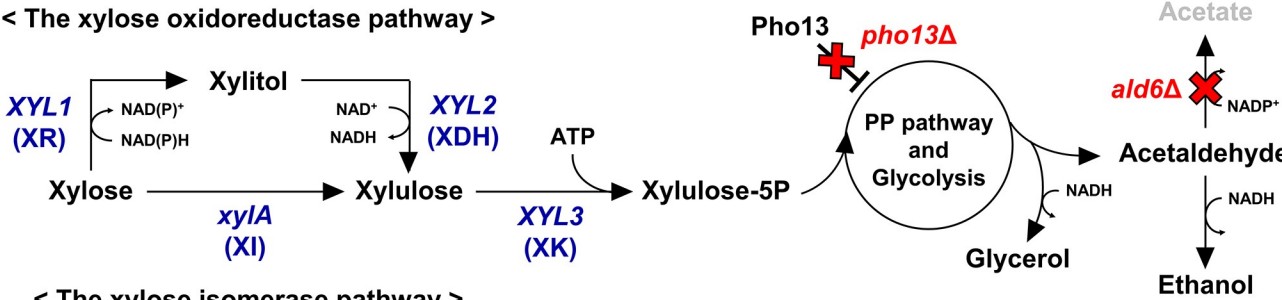

**(B)**

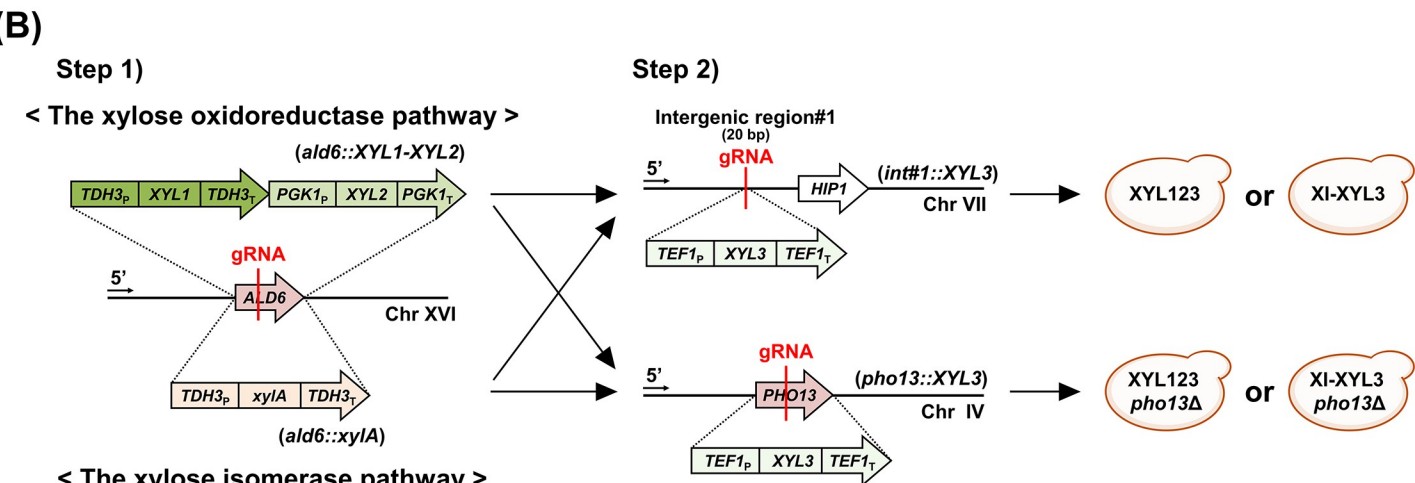

**Fig 1. Construction of isogenic *S. cerevisiae* strains expressing a different type of xylose pathways.** (A) Two different xylose pathways. (B) Strain construction using a precise Cas9-based genome integration strategy.

Adaptive evolution have been the most commonly used and the most effective approach to improve both the strains expressing the oxidoreductase pathway [3,9,10] and the strains expressing the isomerase pathway [11–13]. Some of the evolved strains were subjected to genome sequencing to identify genetic changes responsible for the improved phenotypes. In prior studies, the loss of function mutation of the *PHO13* gene encoding phosphatase with a broad substrate spectrum was identified as a key mutation of an evolved strain expressing the xylose oxidoreductase pathway [3,14,15]. Deletion of the *PHO13* gene (*pho13Δ*) now provides simple, effective, and transferrable to different strain backgrounds expressing the xylose oxido-reductase pathway [3,14–16]. Moreover, it was further confirmed that *pho13Δ* leads to tran-scriptional and metabolic shifts toward efficient xylose fermentation [17,18]. However, it has not been clearly understood how strains expressing the isomerase pathway can be simply improved, although there have been several attempts of genome sequencing of the evolved strains expressing the isomerase pathway [19–24].

In this study, we constructed two isogenic strains expressing either the xylose oxidoreduc-tase pathway or the xylose isomerase pathway through a precise Cas9-based genome integra-tion strategy [25,26]. Deletion of the *PHO13* gene, adaptive evolution, the upregulation of the PP pathway, and some other strategies were performed to identify the most critical and simple factor to improve the strain expressing the xylose isomerase pathway.

## Materials and methods

### Culture conditions

The *S. cerevisiae* strains were routinely grown on yeast extract-peptone (YP) medium (10 g/L yeast extract, 20 g/L peptone) containing 20 g/L of glucose (YPD) at 30˚C for fermentation experiments. The medium used for yeast transformation was YPD agar plate supplemented with antibiotics (100 μg/mL nourseothricin sulfate, 300 μg/mL hygromycin B, 300 μg/mL G418 sulfate). *Escherichia coli* TOP10 (Invitrogen, Carlsbad, CA, USA) was used to amplify plasmid DNA. *E. coli* was cultured in Luria-Bertani (LB) medium (5 g/L yeast extract, 10 g/L tryptone, 10 g/L NaCl) at 37˚C and, if necessary, 100 μg/mL ampicillin (LBA) or 50 μg/mL kanamycin (LBK) was added.

### Plasmid construction and strain engineering

The strains and plasmids used in this study are summarized in Table 1 and S1 Table, respec-tively. The detailed materials and methods for plasmid and strain construction are available in the online supplementary information (S1 Text).

### Flask fermentation experiments

After pre-cultivation in YP medium with 20 g/L of glucose for 24 hours at 250 rpm, yeast cells were harvested by centrifugation at 3,134 ×g, at 4˚C for 5 min, and washed with distilled water. The initial cell concentration was adjusted to an optical density at 600 nm (OD$_{600}$) of 1.0 or 50.0, which corresponds to initial cell density of 0.5 and 25 g DCW/L, respectively, and the cell pellet was inoculated into 20 mL of YP medium containing 40 g/L xylose. Oxygen-lim-ited cultivation was performed at 30˚C in a 100-mL Erlenmeyer flask using a rotary shaker at 80 rpm. Anaerobic cultivation was performed at 30˚C in 125-mL serum bottles using a rotary shaker at 130 rpm. To remove oxygen, the serum bottles were flushed with nitrogen that had passed through a heated, reduced copper column. All experiments were performed in biologi-cal triplicate.

**Table 1. *Saccharomyces cerevisiae* strains used in this study.**

| Strains | Description/relevant genotype | Reference |
|---|---|---|
| D452-2 | *Matα leu2 his3 ura3* | [27] |
| XYL12 | D452-2 *ald6::TDH3_P-XYL1-TDH3_T-PGK1_P-XYL2^{1)}-PGK1_T* | This study |
| XYL123 | XYL12 *int#1::TEF1_P-XYL3-TEF1_T*; DY01 | [25] |
| XYL123 *pho13Δ* | XYL12 *pho13::TEF1_P-XYL3-TEF1_T*; DY02 | [25] |
| XYL123e | XYL123 evolved in 40 g/L xylose | This study |
| XI | D452-2 *ald6::TDH3_P-xylA-TDH3_T* | This study |
| XI-XYL3 | XI *int#1::TEF1_P-XYL3-TEF1_T* | This study |
| XI-XYL3 *pho13Δ* | XI *pho13::TEF1_P-XYL3-TEF1_T* | This study |
| XI-XYL3 *TAL1* | XI-XYL3 with insertion of a *TEF2* promoter in front of *TAL1* gene | This study |
| XI-XYL3e | XI-XYL3 evolved in 100 g/L xylose | This study |
| XI-XYL3 *pho13Δe* | XI-XYL3 *pho13Δ* evolved in 100 g/L xylose | This study |
| XI-XYL3 *gre3Δ* | XI-XYL3 *gre3::KanMX* | This study |
| XI-XYL3 *sor1Δ* | XI-XYL3 with *SOR1* deletion | This study |
| (XI)₂-XYL3 | XI-XYL3 *int#6::TDH3_P-xylA-TDH3_T* | This study |
| XI-(XYL3)₂ | XI-XYL3 *int#9::TEF1_P-XYL3-TEF1_T* | This study |
| (XI)₂-XYL3 *pho13Δ* | XI-XYL3 *int#6::TDH3_P-xylA-TDH3_T* | This study |
| δ(XI)-XYL3 | XI-XYL3 *leu2::LEU2* pYS-δXI | This study |
| δ(XI)-XYL3 *pho13Δ* | XI-XYL3 *leu2::LEU2* pYS-δXI with *PHO13* deletion | This study |

*XYL1*, *XYL2*, and *XYL3* originated from *Pichia stipitis*; *xylA* originated from *Orpinomyces* sp.

## Volumetric growth rate analysis at various xylose concentrations

To compare growth rate after *PHO13* gene deletion, all strains were pre-cultured in 10 mL of YP medium containing 20 g/L of glucose, and the pre-cultured cells were harvested at mid-exponential phase and inoculated into 3 mL of YP medium containing various concentrations of xylose after washing twice with sterilized water. Growth rate analysis was performed in 14-mL Round-Bottom Tubes (SPL, Pocheon, Korea) at 30°C and 250 rpm with a low initial cell density (0.5 g DCW/L). The control (XYL123 and XI-XYL3) and *pho13Δ* (XYL123 *pho13Δ* and XI-XYL3 *pho13Δ*) strains were compared at 1–200 g/L xylose. Volumetric growth rates (g/L-h) were calculated based on the starting and ending points of the exponential phase. All experiments were performed in biological triplicate.

## Transcriptional analysis by RT-qPCR

RT-qPCR was performed by extracting RNA from cells of the exponential phase as previously described [18]. All of the strains were grown in YP media containing 20 g/L glucose or 40 g/L xylose. The cDNA solution, prepared from 1 μg of RNA using the ReverTra Ace® qPCR RT Master Mix (TOYOBO, Osaka, Japan), was used directly with primers and iQ™ SYBR Green Supermix (Bio-Rad, Hercules, CA, USA) for quantitative PCR (qPCR). qPCR was performed using a CFX Connect™ Real-Time PCR Detection System (Bio-Rad, Hercules, CA, USA). The primers used for RT-qPCR are described in S2 Table. All of the measurements were performed in three technical replicates for each biological triplicate.

## Adaptive laboratory evolution

After pre-cultivation in YP medium containing 20 g/L of glucose for 24 hours at 250 rpm, yeast cells were harvested by centrifugation at 15,928 ×g, at 4°C for 1 min. The pre-cultured

cells were washed with distilled water, and the cell pellet was inoculated into 20 mL of YP medium containing 40 g/L or 100 g/L xylose under oxygen-limited conditions (80 rpm). The initial cell densities were adjusted to 0.5 g DCW/L. Growth adaption was performed at 30˚C in a 100-mL Erlenmeyer flask using a rotary shaker at 80 rpm. The cells were transferred to fresh medium when they reached exponential phase. The growth adaption was continued for about 90 days. To confirm that the strains had evolved, three independent colonies were isolated from the YPD agar plate and evaluated by fermentation performances under oxygen-limited conditions (80 rpm).

## HPLC analysis

Quantitation of xylose, xylitol, glycerol, acetate, and ethanol in the culture was analyzed by a high-performance liquid chromatography (HPLC; Agilent Technologies, 1260 series, USA) equipped with a Rezex-ROA Organic Acid H+ (8%) (150 mm × 4.6 mm) column (Phenomenex Inc., Torrance, CA, USA). Columns were eluted with 0.005 N $H_2SO_4$ at 50˚C, and the flow rate was set at 0.6 mL/min, as described previously [28]. Acetate was not detected in all fermentations, and the results were omitted from the figures and tables.

## Intracellular metabolite extraction and derivatization

Metabolite extraction was carried out with some modification of the previously described method [29]. Briefly, 5 mL of cell cultures at mid-exponential growth phase were quenched by quick injection into 25 mL of 60% (v/v) cold methanol (HEPES, 10 mM; pH 7.1) at -40˚C. The cells were centrifuged at 3,134 ×g at -20˚C for 5 min, then discard supernatant thoroughly. Subsequently, 1 mL of 75% (v/v) boiling ethanol (HEPES, 10 mM; pH 7.1) was added to the quenched cell pellet, then make sure that cell pellet should be suspended well with boiling ethanol solution. The mixture was then vortexed for 30 s in a max force, incubated at 80˚C for 5 min. The cell residues were separated from the extract by centrifugation at 15,928 ×g at 4˚C for 1 min. The supernatant was then vacuum-dried for 5 h using a speed vacuum concentrator (Labconco, Kansas City, MO, USA).

The vacuum-dried samples were derivatized by methoxyamination and trimethylsilylation as previously described with some modifications [29]. For methoxyamination, 40 μL of methoxyamine hydrochloride in pyridine (40 mg/mL; Sigma-Aldrich, St. Louis, MO, USA) was added to the samples and incubated at 30˚C for 90 min. For trimethylsilylation, 40 μL of N-methyl-N-(trimethylsilyl)trifluoroacetamide (Sigma-Aldrich, St. Louis, MO, USA) was added to the samples and incubated at 37˚C for 30 min.

## Intracellular metabolite analysis using GC/MS

GC/MS analysis was conducted using an Agilent 6890 GC equipped with an Agilent 5973 MSD as described previously with some modifications [17]. A 1 μL aliquot of derivatized samples was injected into the GC in a split mode (10:1) and separated on an RTX-5Sil MS column (30 m × 0.25 mm, 0.25-μm film thickness; Restek, Bellefonte, PA, USA). The initial oven temperature was set at 75˚C for 1 min, and then ramped at 15˚C/min to a final temperature of 300˚C, held for 2 min. Helium was used as a carrier gas at a constant flow rate of 0.7 mL/min. The temperatures of ion source and transfer line were set at 230˚C and 280˚C, respectively. An electron impact of 70 eV was used for ionization. The mass selective detector was operated in scan mode with a mass range of 50–550 m/z.

## Results

### Construction and comparison of two isogenic strains expressing xylose oxidoreductase pathway or xylose isomerase pathway

Two isogenic strains expressing either a xylose oxidoreductase pathway (*XYL1-XYL2*) or the xylose isomerase pathway (*xylA*) were constructed as follows (Fig 1B). For the origin of the genes, *XYL1* and *XYL2* from yeast *P. stipitis* [28] and *xylA* from anaerobic fungus *Orpinomyces* sp. (GenBank No. MK335957) were used which are known to have the highest catalytic activities among the same group of enzymes tested [30,31]. Because acetaldehyde dehydrogenase encoded by the *ALD6* gene plays a major role in acetate accumulation [32], and because acetate is detrimental to xylose metabolism of the oxidoreductase strains [3] as well as the isomerase strains [33,34], the *ALD6* gene was often selected as knockout target for xylose strains [35,36]. In the present study, therefore, the xylose pathway genes, *XYL1-XYL2* or *xylA*, were genome-integrated by replacing the *ALD6* gene by a Cas9-based genome integration strategy, resulting in the *XYL12* (*ald6::XYL1-XYL2*) and the XI (*ald6::xylA*) strains, respectively. Next, the *XYL3* gene encoding xylulokinase, of which overexpression is required for both pathways, was genome-integrated at an intergenic region (int#1, Fig 1B), resulting in the XYL123 and XI-XYL3 strains, respectively.

When fermenting 40 g/L xylose under oxygen-limited conditions with a low initial cell density (0.5 g DCW/L), the resulting strains showed different phenotypes; while the XYL123 strain consumed over 90% xylose and produced ethanol within 72 h (Fig 2A), the XI-XYL3 strain consumed 10% xylose in the same time period and no ethanol was detected (Fig 2B and Table 2). The difference in the rate of xylose metabolism is primarily due to the thermodynamic advantage of the oxidoreductase pathway compared to the isomerase pathway, as previously reported [37]. Ethanol production by the XI-XYL3 strain was only possible to detect under anaerobic conditions with a high initial cell density (25 g DCW/L) (Table 3). The accumulation of significant amount of xylitol by the XI-XYL3 strain (5.0 g/L) compared to the XYL123 strain (0.6 g/L) was likely due to endogenous non-specific xylose reductase activities (Gre3), which is more significant when the rate of xylose metabolism is slow [38].

### Effects of the *PHO13* deletion on xylose fermentation by two xylose-metabolizing strains

To determine the effects of *PHO13* deletion, the *XYL3* gene was genome-integrated by replacing the *PHO13* gene of the *XYL12* and XI strains, resulting in the XYL123 *pho13Δ* and XI-XYL3 *pho13Δ* strains (Fig 1B). When 40 g/L of xylose was provided, the XYL123 *pho13Δ* strain consumed xylose completely in 48 h, resulting in the fermentation time being reduced by 33%, and the growth rate being increased by 1.54-fold as compared to those of the XYL123 strain (Fig 2C and 2D). In addition, the XYL123 *pho13Δ* strain exhibited 1.76-fold higher specific ethanol productivity and a 6.33-fold increase in xylitol yield as a by-product, than that seen in the XYL123 strain (Fig 2E and Table 2). These results confirmed that *pho13Δ* improved the xylose fermentation rate in a strain expressing a heterologous xylose oxidoreductase pathway as previously described [3,18,43,44]. However, *pho13Δ* did not affect xylose consumption or by-product yields in the XI-XYL3 strain expressing the xylose isomerase pathway (Fig 2F and Table 2). Under anaerobic conditions with a high initial cell density (25 g DCW/L), *pho13Δ* rather decreased ethanol production by 20% (Table 3).

### *PHO13* deletion-induced transcriptional and metabolic changes in two xylose-metabolizing strains

It has been reported that *pho13Δ* induces significant changes at both transcriptional and metabolic levels in the strains expressing the xylose oxidoreductase pathway. First, *pho13Δ* increases

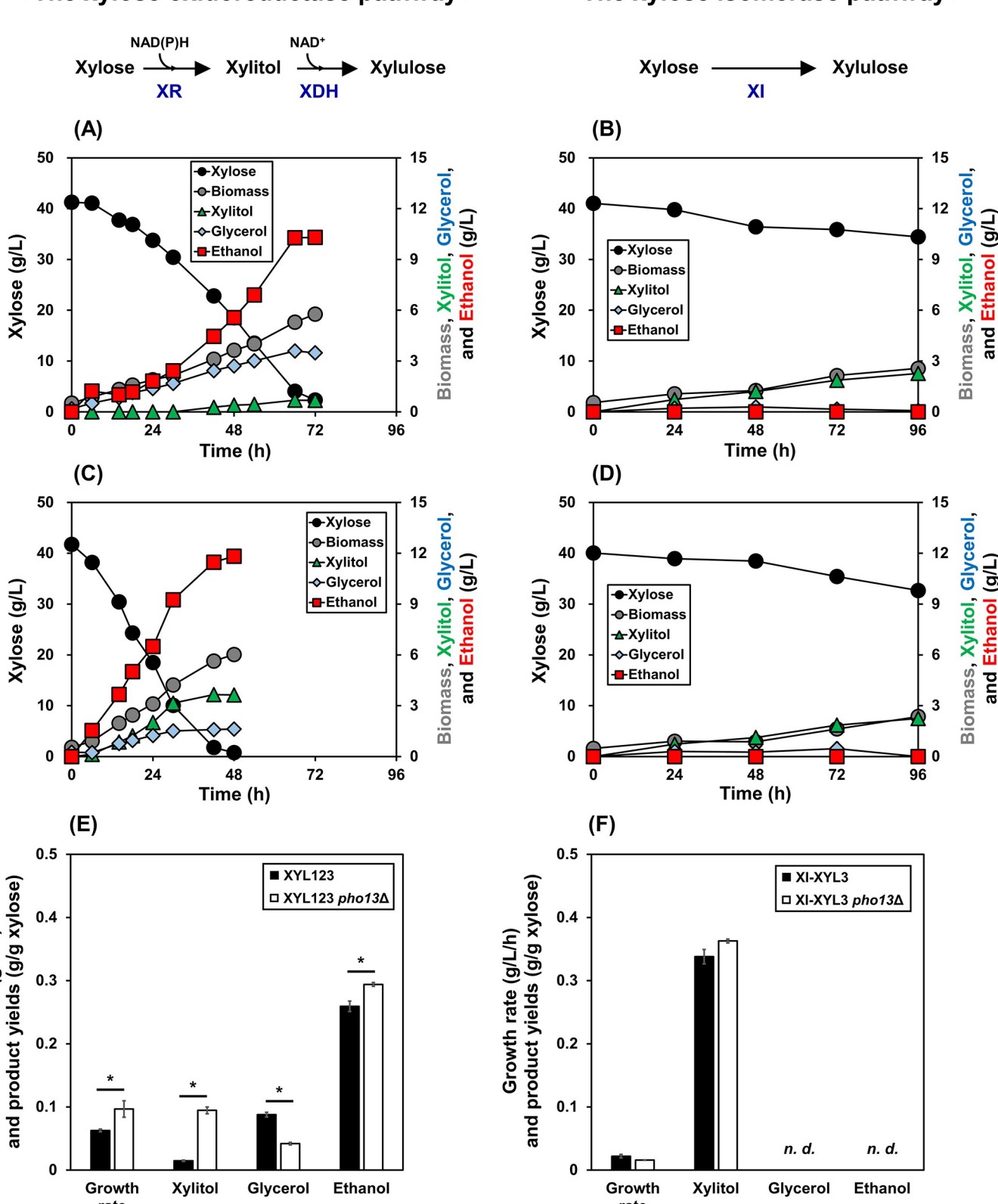

**Fig 2. Effect of *PHO13* deletion on xylose fermentation by two xylose-metabolizing strains.** (A) The XYL123 strain expressing the xylose oxidoreductase pathway and (B) the XI-XYL3 strain expressing the xylose isomerase pathway were compared to their corresponding *pho13*Δ mutants (C and D, respectively).

(E, F) Volumetric growth rates (g/L-h) and product yields (g/g) of the xylose fermentations. Fermentations were performed in YP medium containing 40 g/L xylose under oxygen-limited conditions (80 rpm), with a low initial cell density (0.5 g DCW/L). Asterisks denote statistically significant differences (Student's t-test, $p < 0.05$). *n. d.*; Not detected.

the expression levels of *TAL1*, which encodes sedoheptulose-7-phosphate:D-glyceraldehyde-3-phosphate transaldolase in the PP pathway, under both glucose [18,39] and xylose conditions [18], which was confirmed in the XYL123 strain of the present study as well (Fig 3A). Also, *pho13Δ* leads to the reduction in intracellular sedoheptulose-7-phosphate (S7P) and sedoheptulose levels during xylose metabolism [17], as confirmed in Fig 3B. However, in the XI-XYL3 strain, the *TAL1* activation by *pho13Δ* was observed only under glucose conditions (7.3-fold increase, Fig 3A) but not under xylose conditions. Moreover, S7P was not accumulated in the XI-XYL3 strain during xylose metabolism (Fig 3B). Therefore, it was concluded that *pho13Δ* does not contribute to xylose metabolism of the XI-XYL3 strain neither at the transcription levels nor at the metabolic levels under the conditions we tested. It is hypothesized that some metabolic conditions are required for *PHO13* deletion-induced transcriptional activation of *TAL1*, which is independent from the type of a metabolizing sugar. Because the XI-XYL3 strain metabolizes xylose very slowly, a lack of ATP and/or a low level of some metabolic intermediates could be associated with undesirable conditions for the *TAL1* activation.

## Adaptive evolution of two xylose-metabolizing strains

Concentration of xylose higher than 10 g/L inhibits the growth of *S. cerevisiae* strains expressing a xylose oxidoreductase pathway, which provides driving force for adaptive evolution as described previously [3]. The XYL123 strains constructed in the present study also showed decrease in the growth rates when the xylose concentration exceeded 10 g/L (Fig 4A). Also, when the XYL123 strain was subjected to serial sub-cultures on 40 g/L xylose, gradual increase in the growth rate of the culture was observed, suggesting the progress of adaptive evolution (Fig 4C). In fact, some selected mutants isolated from the evolved cultures showed improved xylose fermentation capabilities that were comparable to the XYL123 *pho13Δ* strain (S1 and S2 Figs and S3 Table). It is interesting to note that the growth of the XYL123 *pho13Δ* strain was

**Table 2. Fermentation profiles of engineered *S. cerevisiae* expressing heterologous xylose fermentation pathways.**

| Strain | Growth rate (g/L-h) | Xylose consumed (g/L) | Xylose consumption rate (g/L-h) | Product titers (g/L) | | | $Y_{Xylitol}$ | $Y_{Glycerol}$ | $Y_{Ethanol}$ | $P_{Ethanol}$* |
|---|---|---|---|---|---|---|---|---|---|---|
| | | | | Xylitol | Glycerol | Ethanol | | | | |
| XYL123 | 0.06 ± 0.00 | 39.9 ± 1.0 | 0.53 ± 0.02 | 0.6 ± 0.2 | 3.3 ± 0.1 | 10.6 ± 0.4 | 0.01 ± 0.00 | 0.09 ± 0.01 | 0.26 ± 0.01 | 0.06 ± 0.01 |
| XYL123 *pho13Δ* | 0.10 ± 0.01 | 41.1 ± 1.4 | 0.93 ± 0.04 | 3.7 ± 0.2 | 1.6 ± 0.4 | 11.9 ± 0.5 | 0.09 ± 0.01 | 0.04 ± 0.01 | 0.29 ± 0.01 | 0.10 ± 0.01 |
| XI-XYL3 | 0.04 ± 0.01 | 14.7 ± 0.5 | 0.06 ± 0.00 | 5.0 ± 0.3 | *n. d.* | *n. d.* | 0.34 ± 0.01 | *n. d.* | *n. d.* | *n. d.* |
| XI-XYL3 *pho13Δ* | 0.03 ± 0.00 | 15.1 ± 1.2 | 0.07 ± 0.00 | 5.8 ± 0.1 | *n. d.* | *n. d.* | 0.36 ± 0.00 | *n. d.* | *n. d.* | *n. d.* |
| (XI)$_2$-XYL3 | 0.08 ± 0.01 | 23.2 ± 1.5 | 0.10 ± 0.01 | 6.7 ± 0.1 | *n. d.* | *n. d.* | 0.29 ± 0.02 | *n. d.* | *n. d.* | *n. d.* |
| (XI)$_2$-XYL3 *pho13Δ* | 0.06 ± 0.01 | 21.0 ± 1.0 | 0.09 ± 0.01 | 7.8 ± 0.4 | *n. d.* | *n. d.* | 0.37 ± 0.00 | *n. d.* | *n. d.* | *n. d.* |
| δ(XI)-XYL3 | 0.10 ± 0.01 | 27.7 ± 0.2 | 0.15 ± 0.01 | 10.5 ± 0.2 | 0.4 ± 0.1 | 1.0 ± 0.0 | 0.34 ± 0.01 | 0.07 ± 0.00 | 0.16 ± 0.01 | < 0.00 |
| δ(XI)-XYL3 *pho13Δ* | 0.11 ± 0.01 | 32.0 ± 0.6 | 0.16 ± 0.00 | 9.2 ± 0.5 | *n. d.* | 1.6 ± 0.0 | 0.29 ± 0.00 | *n. d.* | 0.20 ± 0.01 | < 0.00 |

All strains were cultured in YP medium containing 40 g/L xylose under oxygen-limited conditions (80 rpm) with a low initial cell density (0.5 g DCW/L). All parameters were calculated when either more than 90% of xylose was consumed or fermented for up to 240 h. Acetate was not detected during the xylose fermentation.

Parameters: $Y_{Xylitol}$, Xylitol yield (g xylitol/g xylose); $Y_{Glycerol}$, Glycerol yield (g glycerol/g xylose); $Y_{Ethanol}$, Ethanol yield (g ethanol/g xylose); $P_{Ethanol}$*, Specific ethanol productivity (g/g cell/h); *n. d.*, not detected.

**Table 3. Fermentation profiles of *S. cerevisiae* strains expressing the xylose isomerase pathway derived from *Orpinomyces* sp.**

| Name | Background | Genotype | Conditions | Initial cell (g DCW/L) | Xylose consumed (g/L) | Ethanol titer (g/L) | $\mu_{max}$ | $r_{Xylose}$ | $Y_{Xylitol}$ | $Y_{Ethanol}$ | Reference |
|---|---|---|---|---|---|---|---|---|---|---|---|
| XYL123 pho13Δ | D452-2 | XYL1, XYL2, XYL3, ald6Δ, pho13Δ | AN, YPX (40) | 25.0 | 39.4 | 10.5 | 0.24 | 5.93 | 0.2 | 0.27 | This study |
| XI-XYL3 | D452-2 | xylA*, XYL3, ald6Δ | OL, YPX (40) | 25.0 | 30.3 | - | 0.20 | 0.15 | 0.40 | - | This study |
| | | | AN, YPX (40) | 25.0 | 24.8 | 9.1 | 0.16 | 0.10 | 0.16 | 0.37 | This study |
| XI-XYL3 pho13Δ | D452-2 | xylA*, XYL3, ald6Δ, pho13Δ | OL, YPX (40) | 25.0 | 28.1 | - | 0.20 | 0.13 | 0.42 | - | This study |
| | | | AN, YPX (40) | 25.0 | 19.4 | 7.3 | 0.14 | 0.08 | 0.18 | 0.37 | This study |
| YΔGP/XK/XI | YPH499 | xylA (n = 15), XKS1, gre3Δ, pho13Δ | OL, YPX (50) | 50 g wet cells/L | ~50 | ~22.5 | - | 2.08 | 0.02 | 0.45 | [39] |
| LVY34.4 | LVYA1 | xylA* (n = 36), gre3Δ, RPE1, RKI1, TKL1, TAL1, 2×XKS1, Evolved | O, YPX(30) | 0.25 | < 30.0 | < 13.8 | 0.21 | 1.32 | 0.005 | 0.46 | [22] |
| ADAP8 | INVSc1 | xylA, SUT1, XKS1, Evolved | AN, YPX (20) | 5.0 | 10.8 | 3.4 | 0.13 | 0.09 | 0.26 | 0.32 | [40] |
| YCOA2E | NAPX37 | xylA*, XKS1, HXT7, BGL1, GXS1, Δgre3, Δhxt16, Evolved | OL, YPX (20) | 0.05 | 16.6 | 6.7 | 0.09 | 1.66 | - | 0.41 | [41] |
| O7E15 | NAPX37 | xylA*(n = unknown), XKS1, HXT7, BGL1, GXS1, Δgre3, Δhxt16, Evolved | OL, YPX (40) | 0.2 | 29.7 | 13.0 | - | 0.62 | - | 0.44 | [42] |

Parameters: *xylA**, codon optimized *Orpinomyces* sp. *xylA*, DCW, Dried cell weight; O, Oxygen conditions; OL, Oxygen-limited conditions; AN, Anaerobic conditions; $\mu_{max}$, Volumetric growth rate (g/L-h); $r_{Xylose}$, Xylose consumption rate (g xylose/L/h); $Y_{Xylitol}$, Xylitol yield (g xylitol/g xylose); $Y_{Ethanol}$, Ethanol yield (g ethanol/g xylose).

not inhibited up to 70 g/L xylose (Fig 3A), and it did not undergo adaptive evolutionary process until 65 generations (S3 Fig).

In the XI-XYL3 strain, however, the growth rate gradually increased up to 50 g/L xylose, and there was no initial growth observed at 100 g/L xylose (Fig 4B). At 40 g/L xylose, therefore, serial sub-cultures of the XI-XYL3 strain would not provide high selection pressure for better growers. In fact, until 110 generations, the culture of the XI-XYL3 strain did not show improvement in the growth rates (Fig 4C). At 100 g/L, meanwhile, serial sub-cultures of the XI-XYL3 strain did show slight improvement in the growth rates (Fig 4D); however, the isolated mutants did not have advantages in xylose fermentation (S1 and S4 Figs and S3 Table). Also, it was confirmed again that *pho13Δ* in the XI-XYL3 strain was not as critical as in the XYL123 strain regardless of xylose concentrations (Fig 4B), and during serial subcultures on 40 g/L and 100 g/L xylose (S3 Fig). Therefore, it was confirmed that either *pho13Δ* or evolutionary engineering could be an efficient strategy to improve strains expressing the xylose oxidoreductase pathway; however, the strategy of either *pho13Δ* or evolutionary engineering to improve strain expressing the xylose isomerase pathway did not have more dramatic results than the strain expressing the xylose oxidoreductase pathway.

## Additional copies of *xylA* improves xylose consumption significantly

Several pathway-targeted approaches have been reported for the improvement of the xylose isomerase pathway (Table 3 and Fig 5A). First, the deletion of *GRE3* encoding aldose reductase and/or the deletion of *SOR1* encoding sorbitol (xylitol) dehydrogenase were proposed to

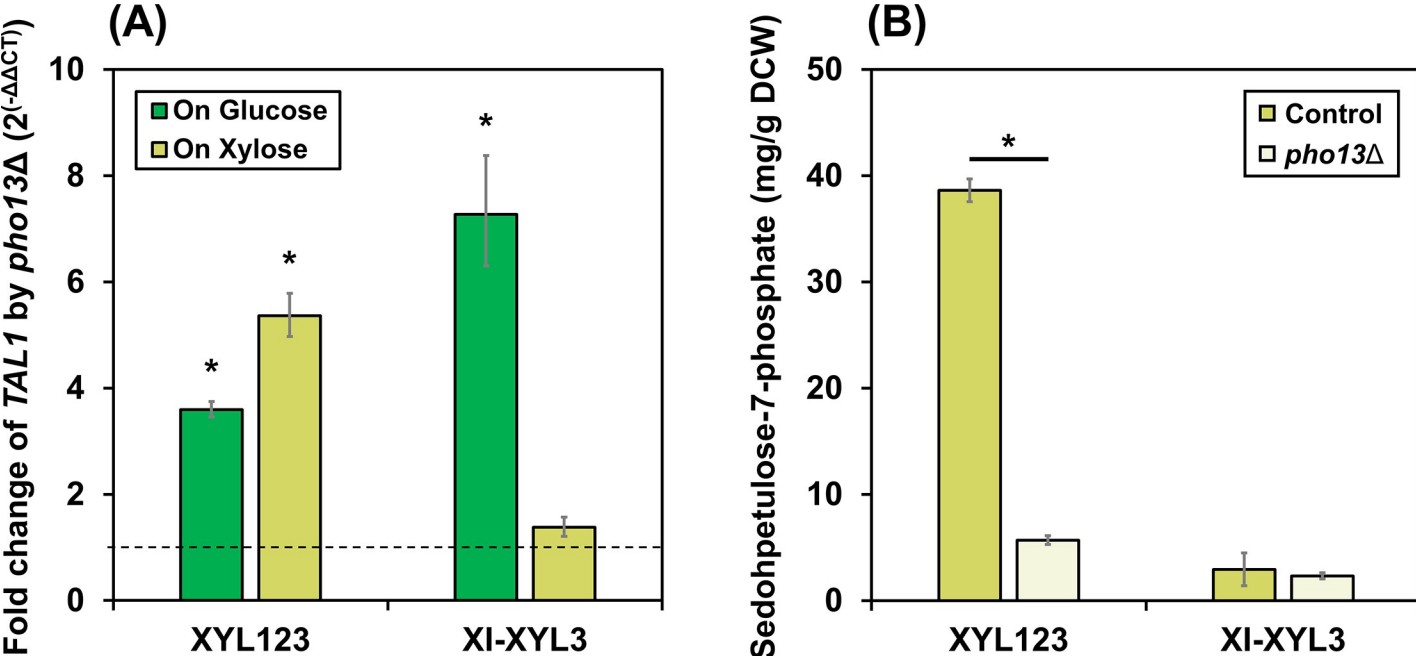

**Fig 3. *PHO13* deletion-induced transcriptional and metabolic changes in two xylose-metabolizing strains.** (A) Fold changes in the mRNA levels of the *TAL1* gene in the *pho13Δ* mutants of the XYL123 and the XI-XYL3 strains growing on glucose or xylose. The dashed line is 1, referring to the *PHO13* wild type. (B) The intracellular concentrations of sedeheptulose-7-phospahte in the *PHO13* wild types (control) and the *pho13Δ* mutants of the XYL123 and the XI-XYL3 strains growing on xylose. Asterisks denote statistically significant differences (Student's t-test, $p < 0.05$).

reduce xylitol accumulation [45,46]. Also, extra copies of *xylA* and/or *XYL3* [22,39,42] were often accompanied with the overexpression of the PP pathway genes such as *TAL1* to improve xylose consumption rates. In Fig 5B, the necessity and contribution of each factors above were evaluated. Although some mutants showed statistically significant increases in growth rate and decreases in xylitol accumulation, none of the single factors contributed to ethanol production from xylose under oxygen-limited conditions (Table 2 and S3 Table). In fact, the most significant improvement was made by the expression of an additional copy of *xylA* in the (XI)₂-XYL3 strain with the xylose consumption rate of 0.10 g/L-h. When multiple copies of the *xylA* gene were integrated at δ sequences (Ty2 transposable element) (S5 Fig), one of the 26 resulting strains (δ(XI)-XYL3) was confirmed for the improved phenotypes (S6 Fig) and for the increased expression levels of the *xylA* gene (35-fold increase, S7 Fig). The δ(XI)-XYL3 strain showed the highest xylose consumption rate (0.15 g/L-h) with detectable amount of ethanol (Fig 5C and Table 2). In addition, with the improved level of xylose consumption, *pho13Δ* was shown to contribute to ethanol yield of the δ(XI)-XYL3 strain while its xylose consumption was not affected (Fig 5C, S6 Fig). However, the xylose consumption rate of the δ(XI)-XYL3 *pho13Δ* strain was still lower than that of the XYL123 *pho13Δ* strain (0.93 g/L-h) as well as those of the previously reported strains with 15–36 copies of the *xylA* gene (1.32–2.08 g/L-h, Table 3) [22,39]. The result suggested that the expression level of the *xylA* gene is one of the most critical factor for efficient xylose consumption, and the δ(XI)-XYL3 strain may have not reached to an optimal level of the *xylA* expression.

## Discussion

There have been numerous attempts to develop *S. cerevisiae* strains fermenting xylose efficiently for decades [37,47,48]. Broadening the substrate range of *S. cerevisiae* is required not

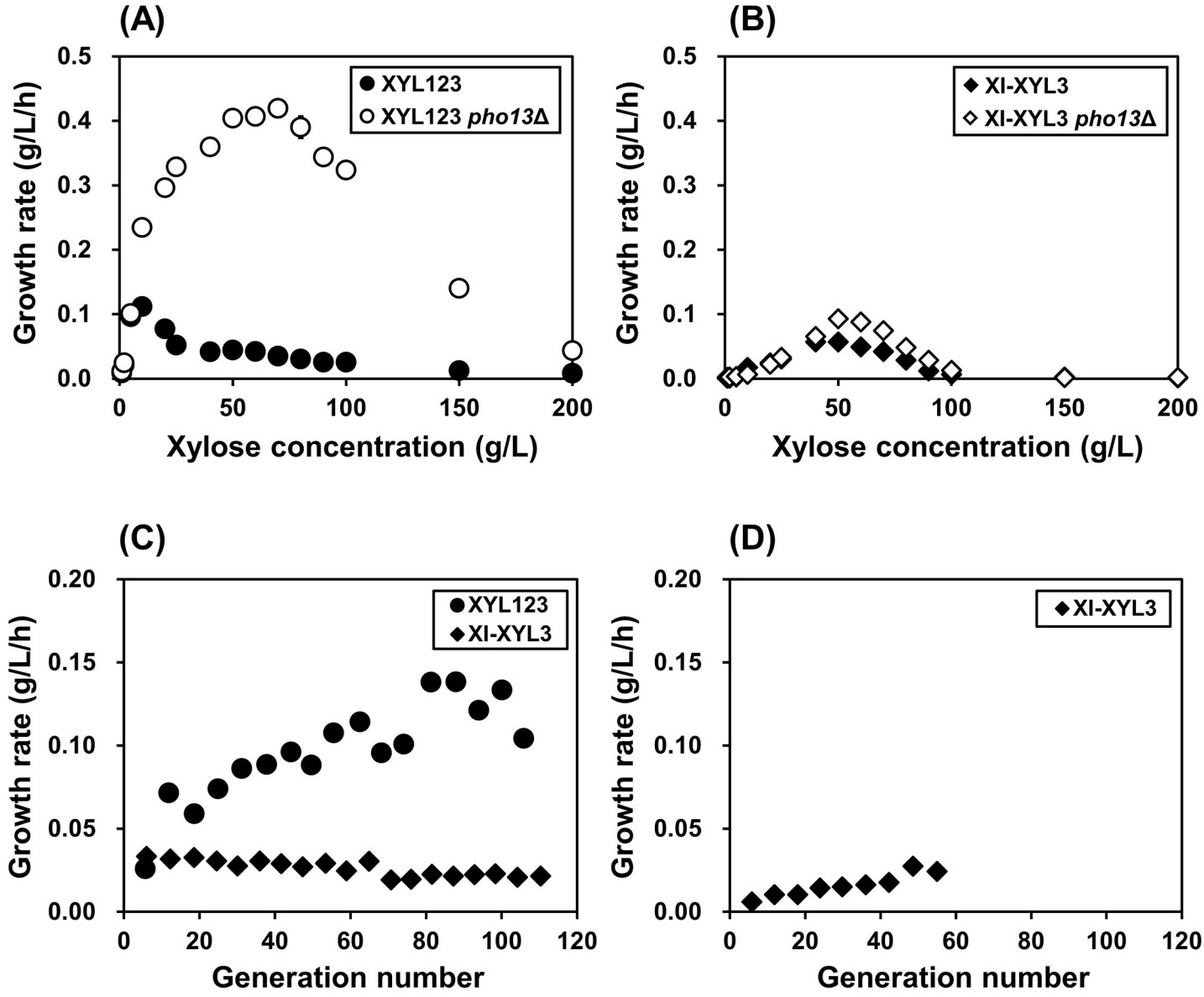

**Fig 4. Adaptive evolution of two xylose-metabolizing strains on xylose.** For adaptive evolution on xylose, growth rates of the XYL123 strains (A) and the XI-XYL3 strains (B) were evaluated under different xylose concentrations. Under growth-liming concentrations of xylose, 40 g/L (C) and 100 g/L (D), the strains were serially subcultured until the described generation numbers.

only to support cellulosic bioprocesses but also to extend product spectrum and efficiency with alternative substrates other than glucose [26,49]. In fact, metabolic engineering for xylose fermentation is a preliminary step toward strain development for desired products. However, the approaches to design efficient *S. cerevisiae* strains expressing the xylose isomerase pathway varied greatly, and adaptive evolution was essential in most prior studies (Table 3) [11,13,20,22,50–52]. It is contradictory to the fact that the optimization of strains expressing the xylose oxidoreductase pathway can be reproducibly achieved by two factors: the constitutive expression of *XYL1*, *XYL2*, and *XYL3* from *P. stipitis* and the deletion of the *PHO13* gene (*pho13Δ*) [3]. Although a prior study presented a reduction in the lag phase by *pho13Δ* in the strain expressing the xylose isomerase pathway, the improvement was not as significant as

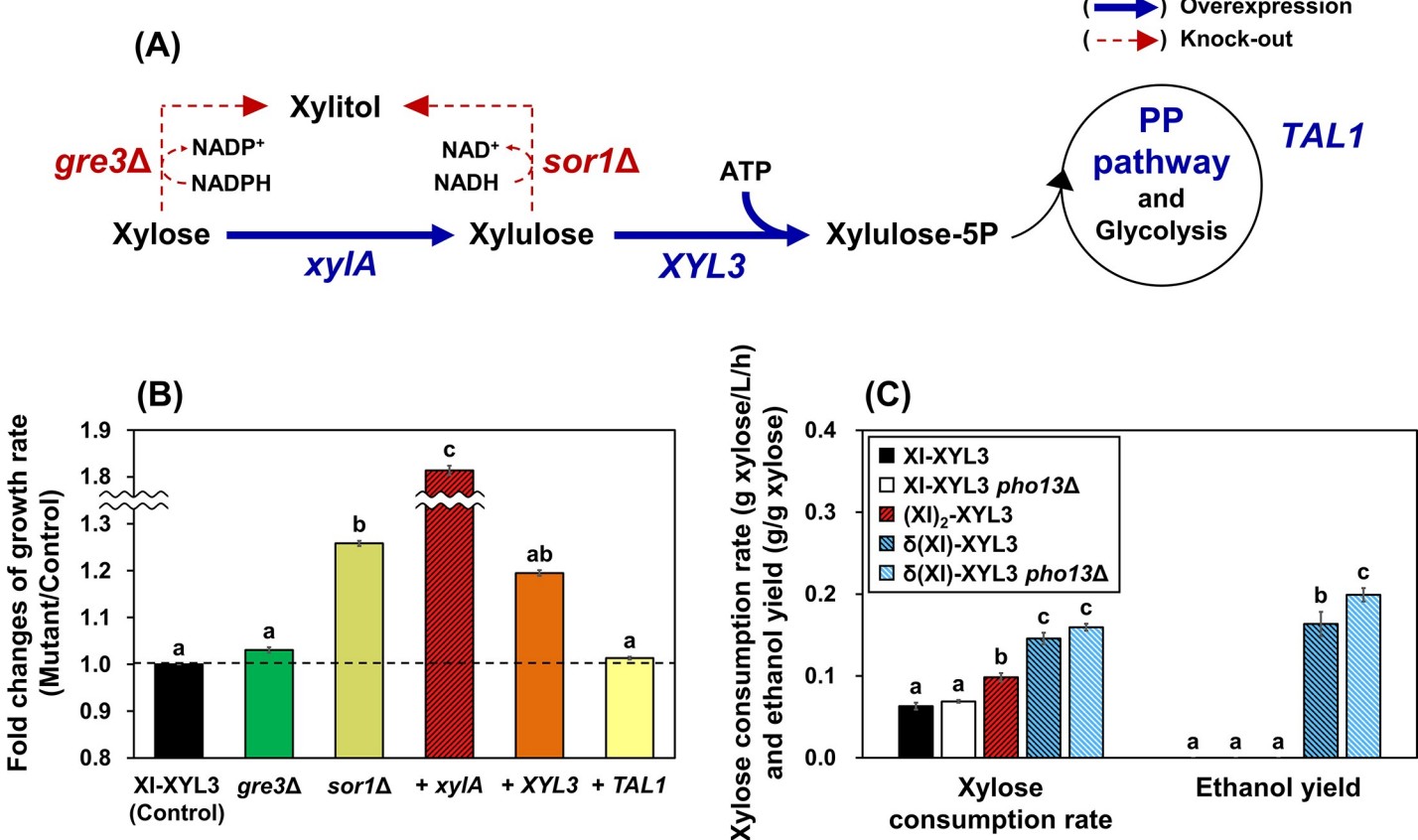

**Fig 5. Pathway-targeted approaches to improve strains expressing the xylose isomerase pathway.** (A) The target genes to be deleted (*gre3Δ*, *sor1Δ*) and the target genes to be overexpressed by integration of a duplicated copy (*xylA*, *XYL3*, *TAL1*). (B) Relative changes in growth rates (g/L-h) on xylose of the engineered strains compared to the XI-XYL3 strain. (C) Comparison of fermentation profiles of the XI-XYL3, (XI)₂-XYL3, and δ(XI)-XYL3 strains, and their *pho13Δ* mutants. All fermentations were performed in YP medium containing 40 g/L xylose under oxygen-limited conditions (80 rpm), with a low initial cell density (0.5 g DCW/L). Different letters represent significant differences across strains within fermentation parameters (Tukey's test, $p < 0.05$). *n. d.*; Not detected.

those achieved by adaptive evolution in the same study [11]. The other study, which reported an 8% increase in the ethanol yield by *pho13Δ* in the xylose isomerase strain, used an extreme condition with an initial OD of 40 [39]. Consistent with previous findings, we also found that *pho13Δ* improved ethanol yield but it was only in the strain expressing multiple copies of *xylA* but not in the single copy *xylA* strain (Fig 5). Also, the xylose consumption rate remained constant in both strains, suggesting the conditional and limited effect of *pho13Δ* in the xylose isomerase strains.

One of the most recent studies reported that nine different expression cassettes of *Piromyces* sp. *xylA*, the overexpression of both *XKS1* and six non-oxidative PP genes (*RPE1*, *RKI1*, *TAL1*, *NQM1*, *TKL1*, *TKL2*), and deletion of the *GRE3* gene are required to construct a xylose-assimilating *S. cerevisiae* strain [50]. The resulting strain was able to produce ethanol after adaptive evolution, in which the loss of function mutation in the *PMR1* gene was critical [50]. In another recent study, two copies of a mutant version of *Piromyces* sp. *xylA* (E15D, E114G, E129D, T142S, A177T, and V433I), the overexpression of *XKS1* and *TAL1*, *pho13Δ* and *GRE3* as well as laboratory evolution were required for xylose fermentation [51,52]. The study concluded that the laboratory evolution was partially contributed by the loss of function mutations in the *PMR1* and *ASC1* genes [51]. Although the metabolic engineering approaches are complicated and different between the two studies, the studies shared the idea that the xylose

isomerase step is the most limiting; therefore, 1) either multiple integration or protein engineering of xylose isomerase is required, and 2) the homeostasis of its inorganic cofactor has to be modified (*pmr1Δ*). The expression of approximately 36 copies of *Orpinomyces sp*. *xylA* [22] and the mutation in *ASK10* for proper folding of isomerase [20] were also proposed to overcome the limitation in xylose isomerase. The above results from recent studies are all consistent with the findings of the present study that the copy number increase in the xylose isomerase gene is the most critical and primarily required (Table 3). However, the optimal level of the copy number of the *xylA* gene varies greatly among studies with the same *xylA* gene derived from *Orpinomyces* sp (Table 3).

It should be noted that the comparison of the two pathways in the present study was limited to the genes originated from *P. stipitis* and *Orpinomyces* sp. for the oxidoreductase and the isomerase pathways, respectively. Considering that the *xylA* gene was originated from strictly anaerobic fungus *Orpinomyces* sp., its functional expression in yeast could have been limited compared to other *xylA* genes originated from bacteria and other fungi [53]. Also, we only compared fermentation properties under oxygen-limited conditions with a low initial cell density. Indeed, under anaerobic conditions with a high initial cell density, where the limited growth of the XI-XYL3 strain can be compensated, the XI-XYL3 strain could produce ethanol at a higher yield (0.37 g/g xylose) than those achieved in the XYL123 strain (0.27 g/g xylose) (Table 3). Nevertheless, engineering an efficient xylose-fermenting strain using the xylose isomerase pathway remains challenging because of the difficulties in reproducing adaptive evolution successfully and achieving optimal copy numbers of the *xylA* gene, as previously reported.

The present study aimed to develop a simple method to optimize *S. cerevisiae* expressing the xylose isomerase pathway: a genome-integrated heterologous xylose isomerase gene (*xylA*) under a strong promoter. We found that adaptive evolution as well as some of the pathway-targeted approaches (*gre3Δ*, *XYL3*, *TAL1*) did not work as efficiently as previously reported. One the other hand, significant improvement in xylose fermentation was achieved by *sor1Δ* as well as multiple integration of the *xylA* gene with or without *pho13Δ*. However, the improved strain was still inferior to an isogenic strain expressing xylose oxidoreductase pathway: xylose reductase (*XYL1*) and xylitol dehydrogenase (*XYL2*). Because the above mentioned approaches for the xylose isomerase pathway were successfully demonstrated in other studies, we think that other unknown factors are required such as different source of the *xylA* gene [53,54], different strain backgrounds [55,56], and/or other metabolic engineering designs. Although recent studies successfully discovered several knockout targets (*ISU1*, *HOG1*, *GRE3*, *IRA2*, *SSK2*) to improve the xylose isomerase pathway, they still required a strain background with the overexpression of the genes in the pentose phosphate pathway and/or the expression of multiple copies of the *xylA* gene [22,24]. With the current level of knowledge regarding xylose isomerase and its functional expression in *S. cerevisiae*, therefore, the xylose oxidoreductase pathway provides a more reproducible strategy to engineer xylose-fermenting strains.

## Supporting information

**S1 Text. Supplementary materials and methods.**
(DOCX)

**S1 Fig.** Growth rate comparison of the evolved colonies of the XYL123 (A), XI-XYL3 (B), and XI-XYL3 *pho13Δ* (C) strains. Two-three most promising colonies were selected from each group, and denoted to XYL123e, XI-XYL3e, and XI-XYL3 *pho13Δ*e, respectively. Strains were cultured in YP medium containing either 40 g/L xylose (A) or 100 g/L xylose (B, C) under oxygen-limited conditions (80 rpm). Volumetric growth rates were calculated at the exponential

phase.
(TIF)

**S2 Fig. Fermentation profiles of the evolved strains expressing the xylose oxidoreductase pathway (the XYL123e strains).** The XYL123 and XYL123 *pho13*Δ strains were used as the controls. Cell density (A), xylose concentrations (B), and fermentation parameters (C) were compared. Fermentations were performed in YP medium containing 40 g/L xylose under oxygen-limited conditions (80 rpm) with a starting $OD_{600}$ of 1.0. Different letters (*a*, *b*, and *c*) represent significant differences ($p < 0.05$, ANOVA method). *n. d.*; Not detected.
(TIF)

**S3 Fig. Adaptive evolution of the *pho13*Δ mutants of the XYL123 and the XI-XYL3 strains on xylose.** Under growth-liming concentrations of xylose, 40 g/L (A) and 100 g/L (A), the strains were serially subcultured until the described generation numbers.
(TIF)

**S4 Fig. Fermentation profiles of the evolved *S. cerevisiae* strains expressing the isomerase pathway on 100 g/L xylose fermentation.** (A, B, C) The XI-XYL3 strain and its evolved strains (XI-XYL3e1, XI-XYL3e2). (C, D, E) The XI-XYL3 *pho13*Δ strain and its evolved strains (XI-XYL3 *pho13*Δe1, XI-XYL3 *pho13*Δe2). The strains were evaluated in YP medium containing 40 g/L xylose under oxygen-limited conditions (80 rpm) with a starting $OD_{600}$ of 1.0. Different letters (*a*, *b*, and *c*) represent significant differences ($p < 0.05$, ANOVA method). *n. d.*; Not detected.
(TIF)

**S5 Fig. Fermentation profiles of 26 mutants overexpressing the *xylA* gene by δ-integration on xylose fermentation.** The XI-XYL3 strain and 26 mutants were evaluated the consumed xylose (g/L) (A) and the produced ethanol (g/L) (B) under oxygen-limited conditions (80 rpm). Six-mutants, which can produce ethanol, were selected and evaluated the xylose consumption rate (g xylose/L/h) and ethanol yield (g ethanol/g xylose) under oxygen-limited conditions (C) and anaerobic conditions (D). Fermentations were performed in YP medium containing 40 g/L xylose, with a starting $OD_{600}$ of 1.0. The dashed line refer to the XI-XYL3 strain.
(TIF)

**S6 Fig. Fermentation profiles of the XI-XYL3, δ(XI)-XYL3 and δ(XI)-XYL3 *pho13*Δ strains on 40 g/L xylose fermentation under two different oxygen conditions.** The strains were evaluated in YP medium containing 40 g/L xylose under oxygen-limited conditions (80 rpm, A-C) and anaerobic condition (D-F) with a starting $OD_{600}$ of 1.0.
(TIF)

**S7 Fig. Comparison of transcriptional levels of *xylA* gene increased by δ-integration in two xylose isomerase pathway strains (XI-XYL3 and δ(XI)-XYL3 strains).** Increased transcriptional levels of the *xylA* gene in the XI-XYL3 and the *xylA* overexpressed strain (δ(XI)-XYL3) by δ-integration was confirmed by RT-qPCR. Fermentations were performed in YP medium containing 40 g/L (YPX40) or 100 g/L (YPX100) xylose, with a starting $OD_{600}$ of 1.0. Asterisks denote statistically significant differences (Student's t-test, $p < 0.05$).
(TIF)

**S1 Table. Plasmids used in this study.**
(DOCX)

**S2 Table. Primers and guide RNAs used in this study.**
(DOCX)

**S3 Table. Fermentation profiles of evolved *S. cerevisiae* expressing the xylose oxidoreductase pathway.**
(DOCX)

## Author Contributions

**Conceptualization:** Deokyeol Jeong, Eun Jung Lee, Soo Rin Kim.

**Data curation:** Deokyeol Jeong.

**Formal analysis:** Deokyeol Jeong, Eun Joong Oh, Soo Rin Kim.

**Funding acquisition:** Eun Jung Lee, Soo Rin Kim.

**Investigation:** Deokyeol Jeong, Eun Jung Lee, Soo Rin Kim.

**Methodology:** Deokyeol Jeong, Soo Rin Kim.

**Resources:** Soo Rin Kim.

**Supervision:** Soo Rin Kim.

**Validation:** Deokyeol Jeong, Eun Joong Oh, Soo Rin Kim.

**Writing – original draft:** Deokyeol Jeong, Eun Joong Oh, Eun Jung Lee, Soo Rin Kim.

**Writing – review & editing:** Deokyeol Jeong, Eun Joong Oh, Ja Kyong Ko, Ju-Ock Nam, Hee-Soo Park, Yong-Su Jin, Eun Jung Lee, Soo Rin Kim.

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
