## [Decision Letter · Decision Letter 0]

7 May 2020

PONE-D-20-07014

Metabolic engineering considerations for the heterologous expression of xylose-catabolic pathways in Saccharomyces cerevisiae

PLOS ONE

Dear Kim,

Thank you for submitting your manuscript to PLOS ONE. After careful consideration, we feel that it has merit but does not fully meet PLOS ONE’s publication criteria as it currently stands. Therefore, we invite you to submit a revised version of the manuscript that addresses the points raised during the review process.

Two of the three reviewer found partial merit in your paper. However, before it could be considered for acceptance in PLOS ONE, major revisions are needed. Plese follow the reviewers' suggestion and reply to all the comments from the three reviewers, with whom I essentially agree.

We would appreciate receiving your revised manuscript by Jun 21 2020 11:59PM. To enhance the reproducibility of your results, we recommend that if applicable you deposit your laboratory protocols in protocols.io, where a protocol can be assigned its own identifier (DOI) such that it can be cited independently in the future. For instructions see: http://journals.plos.org/plosone/s/submission-guidelines#loc-laboratory-protocols

We look forward to receiving your revised manuscript.

Kind regards,

Enrico Baruffini, Ph.D.

Academic Editor

PLOS ONE

Journal Requirements:

Reviewers' comments:

Reviewer's Responses to Questions

**Comments to the Author**

1. Is the manuscript technically sound, and do the data support the conclusions?

Reviewer #1: Yes

Reviewer #2: Yes

Reviewer #3: Partly

2. Has the statistical analysis been performed appropriately and rigorously? 

Reviewer #1: N/A

Reviewer #2: Yes

Reviewer #3: Yes

3. Have the authors made all data underlying the findings in their manuscript fully available?

Reviewer #1: Yes

Reviewer #2: Yes

Reviewer #3: Yes

4. Is the manuscript presented in an intelligible fashion and written in standard English?

Reviewer #1: Yes

Reviewer #2: Yes

Reviewer #3: Yes

5. Review Comments to the Author

Reviewer #1: In this study the authors compare two well-established strategies of generating a xylose fermenting strain of S. cerevisiae – introduction of the oxidoreductase pathway or the xylose isomerase pathway. Establishing a xylose fermenting yeast strain is important because xylose ifs the major non-glucose component of much plant biomass, and it would be useful to be able to turn it into ethanol along with the glucose in a single fermentation reaction.

Various approaches to optimize these pathways have been investigated in the past, this study builds on these observations, and compares them in the two essentially isogenic strains engineered to have the two pathways. The main conclusions are the oxidoreductase pathway responds to changes in Pho13 and Tal1 and to the adaptive evolution protocol implemented, while the isomerase strain responded to increasing the isomerase levels, and that the direct comparison suggests the oxidoreductase pathway may provide the more attractive strategy for generating the xylose fermenting S. cerevisiae strain.

This manuscript seems to fall between a research paper and a review. The majority of the work is essentially repeating a set of previously identified strain modifications in a pair of strains identical except for the added xylose metabolic enzymes; fundamentally consolidating information without providing anything new. The stated goal was to establish the framework for improved production using the isomerase modified strain, but this was not successful – the improvements were limited to increasing levels of the heterologous enzyme. Thus the paper really fails in its ultimate aim, and provides little new in this process. Providing a clear picture of why modifications that work in one framework fail in the other would be useful, but the current format is of very limited value.

Reviewer #2: The authors of the manuscript under review have constructed and compared yeast strains expressing two different pathways for xylose utilization. Some findings reported in the manuscript differ from those reported in earlier studies, and merit a thorough discussion.

Lines 83-86 and 299-303: Prior studies have shown that deleting the PHO13 gene increases xylose uptake and ethanol production in xylose isomerase-expressing strains (for example, Biotechnol Biofuels 2014 7:122; AMB Express 2016 6:4).

Lines 198-200: Details about the integration site should be provided.

Lines 201 ff: Why did XYL123 and XI-XYL3 exhibit different xylose fermentation patterns? This should be discussed.

Lines 233-235: Transcriptional profiling of a pho13 mutant has been shown to upregulate the PP pathway irrespective of the sugar substrate used (Appl Environ Microbiol 2015 81:1601-1609; see Table 4 for TAL1 overexpression). As the results obtained by the authors differ from these findings, this merits a thorough discussion.

Lines 236-240 and 260-267: The authors should discuss why deletion of PHO13 has different effects in strains expressing different xylose assimilation pathways. This is especially important because the results obtained by the authors differ from those reported by other researchers (for example, Biotechnol Biofuels 2014 7:122; AMB Express 2016 6:4). The probable reason for this difference should be discussed.

Lines 288-291 and 321-322: What is the desired expression level of xylA? An earlier report has demonstrated that two copies of a mutant xylose isomerase are sufficient to achieve high xylose consumption and ethanol production rates (Biotechnol Biofuels 2014 7:122).

Lines 323-325: Do the authors expect that different results would be obtained when genes involved in xylose assimilation from other yeast species are overexpressed? Why would that be the case?

Lines 325-328: The interplay of cell density and dissolved oxygen is crucial in determining strain performance. The authors should discuss why the XI-expressing strain performs better under anaerobic conditions.

Lines 341-344: The beneficial effect of PHO13 in XI-expressing strains has been shown to be independent of strain background (Appl Environ Microbiol 2015 81:1601-1609). Expression of xylA from different sources has not been found to affect transcription profiles in yeast (Appl Biochem Biotechnol 2019 189:1007-1019).

Lines 350-352: This is a highly subjective conclusion. The XI pathway is cofactor-independent, and several studies have used the XI pathway to construct efficient xylose-fermmenting strains (for example, BMC Biotechnol 2013 13:110; PLoS Genet 2015 11:e1005010; Front Microbiol 2015 6:1165; Bioresour Bioprocess 2016 3:51).

A note on methods employed: Sufficient details should be provided in the manuscript to enable readers to understand procedures employed without referring to earlier published work. For example, how was cDNA synthesized (line 135)? Which column was used for HPLC analysis (line 155)?

A note on genetic nomenclature: Gene deletions in yeast should be designated by use of lower case letters in italics alone.

Reviewer #3: The manuscript by Jeong et al. compares the use of the oxidoreductase and the isomerase pathways for xylose utilization by S. cerevisiae and concludes that the oxidoreductase pathway provides simpler metabolic engineering strategies. This conclusion is oversimplified and is not supported by the results. This work has sound and interesting experimental results but its presentation and discussion present several flaws and misconceptions. Authors used metabolic engineering strategies previously reported to be successful in XR/XDH engineered strains and confirm its validity for XR/XDH and verify that those are not successful in XI. This should not be surprising as these pathways will affect cell metabolism in a different way. In my opinion, instead of comparing the two pathways directly and choose one that provides simper metabolic engineering strategies, authors should discuss the results obtained for the two pathways and relate those with cell metabolism. One important reference related to this topic is Cunha et al 2019 Biotechnol Biofuels 12(20) that should be introduced in the introduction section and discussed. Some guidelines are given below on how to redirect results discussion.

1) The integration site chosen for both pathways is the ALD6 gene. Are there any evidences that the deletion of this gene affects equally both pathways? As ALD6 codes for a cytosolic aldehyde dehydrogenase that utilizes NADP+ as the preferred coenzyme, besides increasing the flux to ethanol which will consume NADH and produce NAD+, I doubt. As the authors refer in the introduction the XI pathway is cofactor-independent while XR/XDH is cofactor-dependent, presenting a shortage of NAD+. Thus, the comparison might be bias from the very beginning. The decision of deleting ALD6 in the host of the two pathways should be better explained or authors may consider to not delete ALD6.

2) The two strains show completely different physiology on xylose containing media with the strain harbouring the XI pathway presenting xylitol accumulation. Authors attribute the significant accumulation of xylitol by the XI-XYL3 strain to endogeneous enzymes like the non-specific aldose reductase code by GRE3. This has to be better explained as both hosts have the same endogeneous enzymes but only one presents accumulation of xylitol. This could also be related with the point I raised before.

3) On the discussion section, authors state that there is no clear and simple approach to design efficient S. cerevisiae strain with XI pathway. This sentence and the following one, makes no sense as there are several works in literature that use this pathway (XI) together with other modifications. For instance, Cunha et al. 2019 Biotechnol Biofuels 12(20).

4) The authors refer that the copy number increase in xylose isomerase gene is the most critical for XI pathway. What about the enzyme itself? The enzyme efficiency?

5) On page 16, lines 326-327, authors state that under anaerobic conditions with high initial cell density were the XI-XYL3 strain produces ethanol at higher yield than strains with XR/XDH pathway. These comparison makes no sense. Authors compare the XI-XYL3 strain with 25 gDCW/L inoculum with the XYL123 strain with 0.5 gDCW/L.

6) Page 16, lines 329-332: Makes no sense. The XI pathway needs other additional modifications not necessarily the same that function with XR/XDH. Authors should also look into papers that have successful strategies with XI pathway.

7) Pg. 16, line 336, SOR1 deletion has significantly increased growth rate; these results should be better discussed. Again, from the results one may consider that the deletion of ALD6 could be detrimental for XI expression.

8) On M&M, in the section of HPLC analysis, the column and conditions should be given here

9) Pg. 12, Line 243, high? Replace by Concentrations of xylose higher than 10 g/L…

10) Pg. 14, line289, what are these desired expression levels? Some achieved in other studies referred in Table 3 using this xylA gene?

11) Pg. 16 line 345, it is not true as an initial deletion of ALD6 has been made

12) Pg. 17 line 351-352, conclusion not supported by data

13) Table 2, no results on acetate quantification are given, if it was not detected, this information is missing.

14) Table 1, XYL123 is not presented in the table

6. PLOS authors have the option to publish the peer review history of their article (what does this mean?). If published, this will include your full peer review and any attached files.

Reviewer #1: No

Reviewer #2: No

Reviewer #3: No

---

## [Author Response · Author response to Decision Letter 0]

21 Jun 2020

Journal Requirements:

Response: The revised manuscript has been formatted to meet the journal's requirements. 

Response: The ORCID iD has been validated in Editorial Manager.

Reviewer #1

In this study the authors compare two well-established strategies of generating a xylose fermenting strain of S. cerevisiae – introduction of the oxidoreductase pathway or the xylose isomerase pathway. Establishing a xylose fermenting yeast strain is important because xylose is the major non-glucose component of much plant biomass, and it would be useful to be able to turn it into ethanol along with the glucose in a single fermentation reaction.

Various approaches to optimize these pathways have been investigated in the past, this study builds on these observations, and compares them in the two essentially isogenic strains engineered to have the two pathways. The main conclusions are the oxidoreductase pathway responds to changes in Pho13 and Tal1 and to the adaptive evolution protocol implemented, while the isomerase strain responded to increasing the isomerase levels, and that the direct comparison suggests the oxidoreductase pathway may provide the more attractive strategy for generating the xylose fermenting S. cerevisiae strain.

This manuscript seems to fall between a research paper and a review. The majority of the work is essentially repeating a set of previously identified strain modifications in a pair of strains identical except for the added xylose metabolic enzymes; fundamentally consolidating information without providing anything new. The stated goal was to establish the framework for improved production using the isomerase modified strain, but this was not successful – the improvements were limited to increasing levels of the heterologous enzyme. Thus the paper really fails in its ultimate aim, and provides little new in this process. Providing a clear picture of why modifications that work in one framework fail in the other would be useful, but the current format is of very limited value.

Response: We thank the reviewer for the critical evaluation of the value of the present manuscript. In addition to what has been addressed by the reviewer, we would like to highlight that this collective confirmation study can provide a useful standard of the current status of this field, in which there is the overflow of metabolic engineering strategies and their multiple combinations aiming to engineer xylose strains.

Reviewer #2

The authors of the manuscript under review have constructed and compared yeast strains expressing two different pathways for xylose utilization. Some findings reported in the manuscript differ from those reported in earlier studies, and merit a thorough discussion.

Response: We sincerely appreciate the constructive comments from the reviewer. Based on the reviewers’ suggestions, the manuscript was revised, and our point-by-point responses are listed below.

1. Lines 83-86 and 299-303: Prior studies have shown that deleting the PHO13 gene increases xylose uptake and ethanol production in xylose isomerase-expressing strains (for example, Biotechnol Biofuels 2014 7:122; AMB Express 2016 6:4).

Response: We thank the reviewer for pointing out the two important prior studies, which were discussed in the revised manuscript in Lines 379-387.

2. Lines 198-200: Details about the integration site should be provided.

Response: The integration site was added in the revised manuscript (Line 213).

3. Lines 201 ff: Why did XYL123 and XI-XYL3 exhibit different xylose fermentation patterns? This should be discussed.

Response: The following discussion was added to the revised manuscript:

Lines 225-227: The difference in the rate of xylose metabolism is primarily due to the thermodynamic advantage of the oxidoreductase pathway compared to the isomerase pathway, as previously reported [33].

4. Lines 233-235: Transcriptional profiling of a pho13 mutant has been shown to upregulate the PP pathway irrespective of the sugar substrate used (Appl Environ Microbiol 2015 81:1601-1609; see Table 4 for TAL1 overexpression). As the results obtained by the authors differ from these findings, this merits a thorough discussion.

Response: The following discussion was added to the revised manuscript:

Lines 281-286: It is hypothesized that some metabolic conditions are required for PHO13 deletion-induced transcriptional activation of TAL1, which is independent from the type of a metabolizing sugar. Because the XI-XYL3 strain metabolizes xylose very slowly, a lack of ATP and/or a low level of some metabolic intermediates could be associated with undesirable conditions for the TAL1 activation.

5. Lines 236-240 and 260-267: The authors should discuss why deletion of PHO13 has different effects in strains expressing different xylose assimilation pathways. This is especially important because the results obtained by the authors differ from those reported by other researchers (for example, Biotechnol Biofuels 2014 7:122; AMB Express 2016 6:4). The probable reason for this difference should be discussed.

Response: To address the reviewer's concern regarding the different conclusion of the present study, we performed an additional experiment, and finally reached to the same conclusion with others. The detailed results and discussions are as follows:

Lines 348-353: In addition, with the improved level of xylose consumption, pho13∆ was shown to contribute to ethanol yield of the δ(XI)-XYL3 strain while its xylose consumption was not affected (Fig 5C, S6 Fig). However, the xylose consumption rate of the δ(XI)-XYL3 pho13∆ strain was still lower than that of the XYL123 pho13∆ strain (0.93 g/L-h) as well as those of the previously reported strains with 15-36 copies of the xylA gene (1.32-2.08 g/L-h, Table 3) [22,39]. 

Revised Fig 5.

Lines 379-387: Although a prior study presented a reduction in the lag phase by pho13∆ in the strain expressing the xylose isomerase pathway, the improvement was not as significant as those achieved by adaptive evolution in the same study [11]. The other study, which reported an 8% increase in the ethanol yield by pho13∆ in the xylose isomerase strain, used an extreme condition with an initial OD of 40 [39]. Consistent with previous findings, we also found that pho13∆ improved ethanol yield but it was only in the strain expressing multiple copies of xylA but not in the single copy xylA strain (Fig 5). Also, the xylose consumption rate remained constant in both strains, suggesting the conditional and limited effect of pho13∆ in the xylose isomerase strains.

6. Lines 288-291 and 321-322: What is the desired expression level of xylA? An earlier report has demonstrated that two copies of a mutant xylose isomerase are sufficient to achieve high xylose consumption and ethanol production rates (Biotechnol Biofuels 2014 7:122).

Response: We thank the reviewer for pointing out the prior study. Its original work (Appl Environ Microbiol 2012 78:5708) was carefully discussed in Lines 393-395 to address the fact that the desired expression level of xylA varies greatly among studies (2-36 copies), as summarized below:

Line 405-407: However, the optimal level of the copy number of the xylA gene varies greatly among studies with the same xylA gene derived from Orpinomyces sp (Table 3).

7. Lines 323-325: Do the authors expect that different results would be obtained when genes involved in xylose assimilation from other yeast species are overexpressed? Why would that be the case?

Response: Yes. To address what the reviewer pointed out, the following sentence was added in the revised manuscript:

Lines 410-412: Considering that the xylA gene was originated from strictly anaerobic fungus Orpinomyces sp., its functional expression in yeast could have been limited compared to other xylA genes originated from bacteria and other fungi [48]. 

8. Lines 325-328: The interplay of cell density and dissolved oxygen is crucial in determining strain performance. The authors should discuss why the XI-expressing strain performs better under anaerobic conditions.

Response: To address the reviewer’s suggestion, the following sentence was revised in the revised manuscript:

Lines 414-415: Indeed, under anaerobic conditions with a high initial cell density, where the limited growth of the XI-XYL3 strain can be compensated, 

9. Lines 341-344: The beneficial effect of PHO13 in XI-expressing strains has been shown to be independent of strain background (Appl Environ Microbiol 2015 81:1601-1609). Expression of xylA from different sources has not been found to affect transcription profiles in yeast (Appl Biochem Biotechnol 2019 189:1007-1019).

Response: To address the reviewer’s concern, several references were cited to support the following: 

Lines 432-433: we think that other unknown factors are required such as different source of the xylA gene [48,49], different strain backgrounds [50,51],

48. Brat D, Boles E, Wiedemann B (2009) Functional expression of a bacterial xylose isomerase in Saccharomyces cerevisiae. Applied and environmental microbiology 75 8: 2304-2311.

49. Seike T, Kobayashi Y, Sahara T, Ohgiya S, Kamagata Y, et al. (2019) Molecular evolutionary engineering of xylose isomerase to improve its catalytic activity and performance of micro-aerobic glucose/xylose co-fermentation in Saccharomyces cerevisiae. Biotechnology for Biofuels 12: 139.

50. Feng Q, Liu ZL, Weber SA, Li S (2018) Signature pathway expression of xylose utilization in the genetically engineered industrial yeast Saccharomyces cerevisiae. PloS one 13: e0195633.

51. Cunha JT, Soares PO, Romaní A, Thevelein JM, Domingues L (2019) Xylose fermentation efficiency of industrial Saccharomyces cerevisiae yeast with separate or combined xylose reductase/xylitol dehydrogenase and xylose isomerase pathways. Biotechnology for Biofuels 12: 20.

10. Lines 350-352: This is a highly subjective conclusion. The XI pathway is cofactor-independent, and several studies have used the XI pathway to construct efficient xylose-fermmenting strains (for example, BMC Biotechnol 2013 13:110; PLoS Genet 2015 11:e1005010; Front Microbiol 2015 6:1165; Bioresour Bioprocess 2016 3:51).

Response: We agree with the reviewer’s suggestion. Therefore, we revised the conclusion to only highlight the convenience of the strain construction as follows:

Lines 437-439: With the current level of knowledge regarding xylose isomerase and its functional expression in S. cerevisiae, therefore, the xylose oxidoreductase pathway provides a more reproducible strategy to engineer xylose-fermenting strains.

11. A note on methods employed: Sufficient details should be provided in the manuscript to enable readers to understand procedures employed without referring to earlier published work. For example, how was cDNA synthesized (line 135)? 

Response: The manuscript was revised as follows:

Lines 148-149: The cDNA solution, prepared from 1 µg of RNA using the ReverTra Ace® qPCR RT Master Mix (TOYOBO, Osaka, Japan),

12. Which column was used for HPLC analysis (line 155)?

Response: The manuscript was revised as follows:

Lines 170-172: analyzed by a high-performance liquid chromatography (HPLC; Agilent Technologies, 1260 series, USA) equipped with a Rezex-ROA Organic Acid H+ (8%) (150 mm × 4.6 mm) column (Phenomenex Inc., Torrance, CA, USA). Columns were eluted with 0.005 N H2SO4 at 50oC, and the flow rate was set at 0.6 mL/min, as described previously [28]. 

13. A note on genetic nomenclature: Gene deletions in yeast should be designated by use of lower case letters in italics alone.

Response: We thank the reviewer for the suggestion. However, all of the strain names were designated with “pho13∆”; therefore, for consistency throughout the texts, “pho13∆” was used instead of “pho13.”

Reviewer #3

The manuscript by Jeong et al. compares the use of the oxidoreductase and the isomerase pathways for xylose utilization by S. cerevisiae and concludes that the oxidoreductase pathway provides simpler metabolic engineering strategies. This conclusion is oversimplified and is not supported by the results. This work has sound and interesting experimental results but its presentation and discussion present several flaws and misconceptions. Authors used metabolic engineering strategies previously reported to be successful in XR/XDH engineered strains and confirm its validity for XR/XDH and verify that those are not successful in XI. This should not be surprising as these pathways will affect cell metabolism in a different way. In my opinion, instead of comparing the two pathways directly and choose one that provides simper metabolic engineering strategies, authors should discuss the results obtained for the two pathways and relate those with cell metabolism. One important reference related to this topic is Cunha et al 2019 Biotechnol Biofuels 12(20) that should be introduced in the introduction section and discussed. Some guidelines are given below on how to redirect results discussion.

Response: We sincerely appreciate the constructive comments from the reviewer. Based on the reviewers’ suggestions, the manuscript was revised, and our point-by-point responses are listed below.

1) The integration site chosen for both pathways is the ALD6 gene. Are there any evidences that the deletion of this gene affects equally both pathways? As ALD6 codes for a cytosolic aldehyde dehydrogenase that utilizes NADP+ as the preferred coenzyme, besides increasing the flux to ethanol which will consume NADH and produce NAD+, I doubt. As the authors refer in the introduction the XI pathway is cofactor-independent while XR/XDH is cofactor-dependent, presenting a shortage of NAD+. Thus, the comparison might be bias from the very beginning. The decision of deleting ALD6 in the host of the two pathways should be better explained or authors may consider to not delete ALD6.

Response: As the reviewer suggested, a rational of the deletion of the ALD6 gene for both strains was further explained in the revised manuscript as follows:

Lines 211-215: Because acetaldehyde dehydrogenase encoded by the ALD6 gene plays a major role in acetate accumulation [32], and because acetate is detrimental to xylose metabolism of the oxidoreductase strains [3] as well as the isomerase strains [33,34], the ALD6 gene was often selected as knockout target for xylose strains [35,36]. 

2) The two strains show completely different physiology on xylose containing media with the strain harbouring the XI pathway presenting xylitol accumulation. Authors attribute the significant accumulation of xylitol by the XI-XYL3 strain to endogeneous enzymes like the non-specific aldose reductase code by GRE3. This has to be better explained as both hosts have the same endogeneous enzymes but only one presents accumulation of xylitol. This could also be related with the point I raised before.

Response: As the reviewer suggested, xylitol production by the XI-XYL3 strain was further explained as follows:

Lines 228-231: The accumulation of significant amount of xylitol by the XI-XYL3 strain (5.0 g/L) compared to the XYL123 strain (0.6 g/L) was likely due to endogenous non-specific xylose reductase activities (Gre3), which is more significant when the rate of xylose metabolism is slow [38].

3) On the discussion section, authors state that there is no clear and simple approach to design efficient S. cerevisiae strain with XI pathway. This sentence and the following one, makes no sense as there are several works in literature that use this pathway (XI) together with other modifications. For instance, Cunha et al. 2019 Biotechnol Biofuels 12(20).

Response: We thank the reviewer for the critical comment. The above-mentioned reference has been discussed for the possible contribution of industrial strain background on xylose fermentation capability in Lines 430-431 of the revised manuscript. Also, the discussion section has been revised as follows:

Lines 374-377: However, the approaches to design efficient S. cerevisiae strains expressing the xylose isomerase pathway varied greatly, and adaptive evolution was essential in most prior studies [11,13,20,22,50,51,52].

4) The authors refer that the copy number increase in xylose isomerase gene is the most critical for XI pathway. What about the enzyme itself? The enzyme efficiency?

Response: We agree with the reviewer that the origin of xylose isomerase gene and its functional expression are also critical factors for efficient xylose fermentation. However, Table 3 simply demonstrates that xylose consumption rates (rxylose) of the strains with the xylA gene derived from Orpinomyces sp. can vary greatly depending on the copy number of the genes. To address this point concisely, the following sentence was added in the revised manuscript.

Lines 405-407: However, the optimal level of the copy number of the xylA gene varies greatly among studies with the same xylA gene derived from Orpinomyces sp (Table 3).

5) On page 16, lines 326-327, authors state that under anaerobic conditions with high initial cell density were the XI-XYL3 strain produces ethanol at higher yield than strains with XR/XDH pathway. These comparison makes no sense. Authors compare the XI-XYL3 strain with 25 gDCW/L inoculum with the XYL123 strain with 0.5 gDCW/L.

Response: We agree with the reviewer that the comparison was unfair. To address the issue, we performed an additional experiment with the XYL123 pho13∆ strain with a high OD. The result was added to Table 3 in the revised manuscript, and the numbers were modified in Line 415-416.

6) Page 16, lines 329-332: Makes no sense. The XI pathway needs other additional modifications not necessarily the same that function with XR/XDH. Authors should also look into papers that have successful strategies with XI pathway.

Response: Based on the reviewer's suggestion, the sentence was removed. Instead, we specified the difficulties in improving the strains expressing a xylose isomerase pathway. 

Lines 417-420: Nevertheless, engineering an efficient xylose-fermenting strain using the xylose isomerase pathway remains challenging because of the difficulties in reproducing adaptive evolution successfully and achieving optimal copy numbers of the xylA gene, as previously reported. 

7) Pg. 16, line 336, SOR1 deletion has significantly increased growth rate; these results should be better discussed. Again, from the results one may consider that the deletion of ALD6 could be detrimental for XI expression.

Response: As reported previously [36], ald6Δ is not detrimental to a strain expressing the xylose isomerase pathway, and it has been described in Lines 209-213 of the revised manuscript. Based on the reviewer's suggestion, we revised the discussion part of sor1Δ as follows: 

Lines 425-426: One the other hand, significant improvement in xylose fermentation was achieved by sor1Δ as well as multiple integration of the xylA gene with or without pho13Δ.

36. Zhang Y, Lane S, Chen J-M, Hammer SK, Luttinger J, et al. (2019) Xylose utilization stimulates mitochondrial production of isobutanol and 2-methyl-1-butanol in Saccharomyces cerevisiae. Biotechnology for Biofuels 12: 223.

8) On M&M, in the section of HPLC analysis, the column and conditions should be given here

Response: The manuscript was revised as follows:

Lines 170-172: analyzed by a high-performance liquid chromatography (HPLC; Agilent Technologies, 1260 series, USA) equipped with a Rezex-ROA Organic Acid H+ (8%) (150 mm × 4.6 mm) column (Phenomenex Inc., Torrance, CA, USA). Columns were eluted with 0.005 N H2SO4 at 50oC, and the flow rate was set at 0.6 mL/min, as described previously [28]. 

9) Pg. 12, Line 243, high? Replace by Concentrations of xylose higher than 10 g/L…

Response: The manuscript was revised as the reviewer suggested (Line 298).

10) Pg. 14, line289, what are these desired expression levels? Some achieved in other studies referred in Table 3 using this xylA gene?

Response: Yes. For clarity, the text has been revised as follows:

Lines 170-172: However, the xylose consumption rate of the δ(XI)-XYL3 strain was still lower than that of the XYL123 pho13∆ strain (0.93 g/L-h) as well as those of the previously reported strains with 15-36 copies of the xylA gene (1.32-2.08 g/L-h, Table 3) [22,39]. The result suggested that the expression level of the xylA gene is one of the most critical factor for efficient xylose consumption, and the δ(XI)-XYL3 strain may have not reached to an optimal level of the xylA expression.

11) Pg. 16 line 345, it is not true as an initial deletion of ALD6 has been made

Response: We appreciate the reviewer's critical comment. The sentence was removed from the revised manuscript.

12) Pg. 17 line 351-352, conclusion not supported by data

Response: We appreciate the reviewer's critical comment. The conclusion has been revised as follows:

Lines 435-438: With the current level of knowledge regarding xylose isomerase and its functional expression in S. cerevisiae, therefore, the xylose oxidoreductase pathway provides a more reproducible strategy to engineer xylose-fermenting strains.

13) Table 2, no results on acetate quantification are given, if it was not detected, this information is missing.

Response: Yes, acetate was not detected, as described below: 

Lines 172-173: Acetate was not detected in all fermentations, and the results were omitted from the figures and tables.

14) Table 1, XYL123 is not presented in the table 

Response: Table 1 was revised for clarity.

---

## [Decision Letter · Decision Letter 1]

6 Jul 2020

Metabolic engineering considerations for the heterologous expression of xylose-catabolic pathways in Saccharomyces cerevisiae

PONE-D-20-07014R1

Dear Dr. Kim,

We’re pleased to inform you that your manuscript has been judged scientifically suitable for publication and will be formally accepted for publication once it meets all outstanding technical requirements.

Kind regards,

Enrico Baruffini, Ph.D.

Academic Editor

PLOS ONE

Additional Editor Comments (optional):

After the first revision, the reviewers found merit in your work, which can now be accepted by PLOS ONE.

Reviewers' comments:

Reviewer's Responses to Questions

**Comments to the Author**

1. If the authors have adequately addressed your comments raised in a previous round of review and you feel that this manuscript is now acceptable for publication, you may indicate that here to bypass the “Comments to the Author” section, enter your conflict of interest statement in the “Confidential to Editor” section, and submit your "Accept" recommendation.

Reviewer #1: All comments have been addressed

Reviewer #2: All comments have been addressed

2. Is the manuscript technically sound, and do the data support the conclusions?

Reviewer #1: Yes

Reviewer #2: Yes

3. Has the statistical analysis been performed appropriately and rigorously? 

Reviewer #1: N/A

Reviewer #2: Yes

4. Have the authors made all data underlying the findings in their manuscript fully available?

Reviewer #1: Yes

Reviewer #2: Yes

5. Is the manuscript presented in an intelligible fashion and written in standard English?

Reviewer #1: Yes

Reviewer #2: Yes

6. Review Comments to the Author

Reviewer #1: (No Response)

Reviewer #2: The authors have adequately addressed the comments raised on the earlier manuscript, and have revised the manuscript accordingly.

7. PLOS authors have the option to publish the peer review history of their article (what does this mean?). If published, this will include your full peer review and any attached files.

Reviewer #1: No

Reviewer #2: No

---

## [Editor Report · Acceptance letter]

16 Jul 2020

PONE-D-20-07014R1 

Metabolic engineering considerations for the heterologous expression of xylose-catabolic pathways in *Saccharomyces cerevisiae*

Dear Dr. Kim:

I'm pleased to inform you that your manuscript has been deemed suitable for publication in PLOS ONE. Congratulations! Your manuscript is now with our production department. 

Kind regards, 

on behalf of

Dr. Enrico Baruffini 

Academic Editor

PLOS ONE